# Animal Cellulose with Hierarchical Structure Isolated from *Halocynthia aurantium* Tunic as the Basis for High-Performance Pressure-Resistant Nanofiltration Membrane

**DOI:** 10.3390/membranes12100975

**Published:** 2022-10-06

**Authors:** Svetlana V. Kononova, Albert K. Khripunov, Vladislav N. Romanov, Anton S. Orekhov, Alexey A. Mikhutkin, Elena N. Vlasova, Maxim S. Lukasov, Vera V. Klechkovskaya

**Affiliations:** 1Institute of Macromolecular Compounds, Russian Academy of Science, Bolshoy Pr. 31, 199004 St.-Petersburg, Russia; 2Zoological Institute, Russian Academy of Sciences, Universitetskaya Emb., 1, 199034 St.-Petersburg, Russia; 3A.V. Shubnikov Institute of Crystallography FSRC “Crystallography and Photonics” RAS, Leninsky Prospekt 59, 119333 Moscow, Russia; 4National Research Centre “Kurchatov Institute”, pl. Kurchatova, 1, 123098 Moscow, Russia

**Keywords:** ascidian cellulose, tunicin, membrane, structure and transport properties, 3D surface model based on the photogrammetric approach

## Abstract

The structure and transport properties of the new Cellokon-AKH membrane based on animal cellulose obtained from tunic of ascidian *Halocynthia aurantium* were studied. The results of scanning electron microscopy (SEM), FTIR spectroscopy, and the X-ray diffraction data revealed significant differences in the structure and morphology of upper and lower surfaces of this layered film membrane based on animal cellulose. It was shown that the membrane surface is a network of intertwined cellulose fibers, with both denser and looser areas present on the lower surface compared to the completely uniform morphology of the main part of the upper surface. The hierarchical structure of tunicin-based outgrowths evenly distributed over the upper surface was determined and analyzed. The 3D visual representation of the surface structure was performed with the surface reconstruction technique using scanning electron microscope images. A surface model was calculated from the aligned images based on the photogrammetric approach. The transport properties of samples with different prehistory with respect to ethanol, water, and their mixtures of different compositions were studied depending on the pressure. Representing an alcohol-containing gel film in its original state, as solvents are removed, the membrane transforms into a low-permeability fibrillary organized selective film. The obtained results confirmed the possibility of using Cellokon-AKH (dried form) for the filtration of substances with a molecular weight of more than 600 Da in various media. Further study of this new material will allow to get closer to understanding the structure of the studied seabed inhabitants and to use these natural resources more efficiently.

## 1. Introduction

The most important step towards resolving the rapidly growing biospheric crisis is a more rational use of renewable resources [1,2,3,4,5,6]. Cellulose is one of the main biopolymers on the planet with the widest range of practical applications. It has been known for decades that cellulose biosynthesis is carried out not only by plants, but also by other organisms [7,8]. The study of the structure and properties of celluloses of various origins (*Plant cellulose* (PC), *Algal cellulose* (AC), *Bacterial cellulose* (BC) and *Animal cellulose* (AnC)) showed that the celluloses of the early periods of the biosphere in terms of their high-molecular organization can be more useful than PC in solving new problems in technology, medicine, and other areas. Separate interests to researchers are the renewable resources of seas and oceans, which are still scantily explored. This area is now developing rapidly, and, thus, the possibilities of using renewable resources are expanding.

Polysaccharides, including cellulose, and their derivatives are widely used in membrane preparation. Cellulose-based membranes are obtained by various methods; however, the majority of these methods involve the dissolution of cellulose, which is a major problem in itself [9]. To overcome this problem, cellulose derivatives (e.g., esters) are synthesized; then these derivatives are subjected to the corresponding treatment (in the case of esters, hydrolysis) to produce cellulose (for example, hydrolyzed cellulose) [10].

As is known, membranes effective in liquid-phase pressure-driven processes such as nanofiltration should possess a set of specific properties that are directly related to their structure. These properties include the ability to cut off nano-sized particles under pressure filtration [11,12,13]. For this reason, the production of membranes of this type from cellulose leads to the multistage formation of complex architecture films. 

The patent [14] describes the preparation method for a nanofiltration-type membrane that consists of modification of a pre-formed ultrafiltration (UF) membrane; this modification yields a nanofiltration-type composite membrane (the pore size allows the filtration of molecules with molecular masses exceeding 500 Da). Several methods for forming a cellulose-based UV membrane are considered, for example: (i) applying cellulose ester to a synthetic polymer substrate followed by its hydrolysis to form a cellulose layer, (ii) forming a UV film by phase inversion from cellulose solutions in special solvents. The next stage consists of modification of the UF membrane with a crosslinking agent, which results in obtaining the required pore system. The description of the synthesis process demonstrates that the preparation of nanofiltration membranes is a complex and costly process. 

Several attempts have been made to control the properties of hydrolyzed cellulose nanofiltration membranes by introducing nanosized fillers into the polymer matrix [15]. The preparation methods for nanofiltration membranes that involve introducing cellulose crystals into matrices of other polymers have also been described [16,17,18]. These techniques are supposed to use the particular properties of cellulose crystals, i.e., their hydrophilicity combined with low swelling ability in various liquids. The authors of [19] discuss polymeric membranes and methods for their manufacture using nanocrystalline cellulose obtained from wood, cotton, Tunicin, Cladophora sp., Valonia, bacteria, chitin, potato starch, and various combinations of cellulose samples. Surface modification of cellulose is offered either by physical methods (flame or corona discharge) or by a wide range of chemical methods. The proposed membranes, designed using four types of cellulose (including animal), chitin, and starch, are extremely difficult to manufacture. This is also true for multilayer membranes often prepared for use in nanofiltration [20,21].

The cellulose obtained from the tunic of sea squirts (ascidians) has been used as a material for membranes, in particular, membranes for regenerative medicine [22]. The development of such membranes was based on the supposed ability of the produced materials to selectively transfer biologically active substances. In our previous studies, it has been proposed to prepare nanoporous membranes of complex architecture on the basis of tunicin, the material of the mantle (tunic) of *Halocynthia aurantium* (sea peach) [23]. 

*Halocynthia aurantium* is a large (up to 25 cm in height) solitary ascidian. Its body resembles a two-necked jug: food and oxygen enter with water through the inlet siphon, while water removed from the ascidian body through the excretory siphon carries away unnecessary substances. Sea peaches are common in all Far Eastern and some Arctic seas at depths from 4 to 400 m. The Halocynthia body is elastic in water: it is firmly attached to stones and pebbles with root-like outgrowths located on its sole (Figure 1) [24].

Figure 2 shows a schematic diagram of this ascidian, which describes its organs and body structure (proposed by the authors of [25]). The body of ascidians is covered with a tunic, which has a complex structure. On the outside, it is covered with a thin but hard cuticle; under this cuticle is a layer of cells containing tunicin. This is the only example of the formation of a fiber-like substance in an animal body. Under the tunic, a skin–muscular sac is found consisting of a single-layer epithelium and transverse and longitudinal muscle sacs fused with it.

The main chemical components of tunicates are alkaloids and peptides [26,27]. Imidazole alkaloids with antitumor activity were obtained from ascidians. A biologically active food supplement “HAURANTINUM” is obtained from the tunic of *Halocynthia aurantium* in the form of alcohol extract. It was shown that “HAURANTINUM” contains biologically active substances of various natures: more than 15 free amino acids, phospholipids, fatty acids, neutral lipids, prostaglandins, and a number of micro and macro elements, including a large amount of vanadium [28].

Microorganisms associated with invertebrate hosts—ascidians—have also been recognized as a source of bioactive metabolites. It has been established that invertebrate microorganisms producing bioactive metabolites are of great importance for drug discovery and development [25,29,30]. These marine organisms are also considered rich sources of cellulose, the amount of which varies between species [25]. 

The experiments have shown that ascidian tunicin yields 22–73% of polysaccharides (in glucose equivalent) with respect to its dry weight, depending on the type of tunic. In the case of *Halocynthia aurantium*, the maximum percentage of sugars derived from whole tunic was 73% [31], while in the experiments with Pyura and Onemidocarpa ascidians, the yield was equal to 50% [32]. Thus, seemingly unusual polysaccharides, such as tunicin, have been extracted from tunicates. It was revealed that tunicin (marine cellulose of animal origin) in some cases was a highly crystalline substance [33]. Naturally, the degree of crystallinity of cellulose isolated from ascidians depends on the animal species [34]. 

To summarize, tunicates contain and produce a wide range of marine natural products possessing various biologically active properties (Table 1 [25]). For this reason, large amounts of tunicates are harvested from seabed habitats; useful chemical compounds and materials are extracted from these animals.

Considering the availability of this natural raw material containing cellulose, we proposed a membrane from *Halocynthia aurantium* tunicin [23]. The main idea of this work was the use of ascidian tunic cellulose after the isolation of other bioactive substances from this natural material [25,35,36,37].

The technical task and the positive result of the solution proposed in the patent was to manufacture a nanofiltration membrane “Cellokon-AKH” from an available natural source—the ascidian *Halocynthia aurantium*. A simple, environmentally friendly, and economical way to prepare the membrane has been developed [23]. The scientific goal of this work was to optimize the cellulose isolation method in order to remove as many organic compounds as possible from the ascidian mantle, leaving only cellulose. It was necessary to preserve the supramolecular structure inherent in the cellulose of the mantle, since it is this structure that helps the ascidian maintain its integrity in deep water, that is, under conditions of high pressure. 

Another important task was a comprehensive study of the structure and transport properties of the developed membrane based on AnC, the first results of which are presented in this paper. This is important not so much for describing the properties of a specific membrane as for studying the previously undescribed structure of the cellulose backbone of the *Halocynthia aurantium* tunic and the possibilities of its use for the membrane formation. The elements of a complex hierarchical membrane structure were studied using high-resolution SEM. It is important to understand the peculiarities of the natural structure of the ascidian mantle that provide structural stability of the tunicin-based film under increased pressure. In connection with the analysis of the structural features of the membrane, of particular interest is the construction of a 3D surface model of the membrane surface, as well as quantitative analysis of the surface morphology.

## 2. Experimental Section

### 2.1. Materials

The raw material (tunic of the ascidian *Halocynthia aurantium* of the White Sea) was provided by V.N. Romanov (Zoological Institute, Russian Academy of Sciences, St. Petersburg, Russia). The material was pre-prepared according to the method described in [28]. The biomaterial isolated from the animal was immersed in formalin for long-term storage. Before use, the ascidian tunic material was washed many times with water, then with ethanol, after which it was immersed in a medium of 96% ethanol. 

All other reagents were of analytical grade and were obtained from Sigma-Aldrich (St. Louis, MO, USA).

### 2.2. Membrane Preparation

To isolate cellulose from the biomaterial, the mild method of processing natural material was used. The biomaterial was pre-treated by successive extraction with acetone and ethanol; the purpose of such treatment was the release of biologically active substances. The main task at this stage was to preserve the structural integrity of the native mantle of the ascidian *Halocynthia aurantium* with careful removal of low-molecular-weight substances. At the next stage, the obtained samples were kept for 1 h in 0.5 N. aqueous sodium hydroxide solution and 24 h in 0.25 wt% sodium hydroxide solution at 100 °C. The resulting material was thoroughly washed with water until neutral reaction, and then with ethanol.

The resulting porous anisotropic film of complex architecture as a membrane for pressure-driven processes of the liquid separation was used [23]. The membranes were stored under different conditions: (i) in 96 % ethanol; (ii) in the dry state. For this purpose, the samples were preliminarily dried in air under normal conditions.

### 2.3. Membrane Characterization (Methods)

#### 2.3.1. X-ray Diffraction Analysis

X-ray phase studies were performed with a DRON-3M diffractometer (Bourevestnik Inc., Saint Petersburg, Russia) in the reflection mode (Bragg–Brentano geometry) with CuKα radiation (average wavelength λ = 1.54183 Å, Nickel β-filter). The diffractograms were recorded in the continuous mode in the 2θ angle range from 5 to 50 deg; the angular detector speed was equal to 1 grad⋅min^−1^.

#### 2.3.2. IR Spectroscopy Studies

The Cellokon-AKH samples were examined using IR spectroscopy. The spectra were recorded using a Bruker Vertex 70 IR-Fourier spectrometer with a resolution of 4 cm^−1^, the number of scans was 60. The spectra were obtained using a single frustrated total internal reflection (ATR-FTIR) “Pike” micro-attachment with a ZnSe working element. When registering the ATR spectra, a correction was introduced that considers the penetration depth depending on the wavelength. 

#### 2.3.3. Visualization of Samples at Different Angles

Images of upper and lower surfaces of the Cellokon-AKH samples were taken. The characteristics of the membrane were studied before and after filtration. For this, images of two fragments of the membrane were obtained. One sample was a cut out round film fragment already used in filtration experiments (sample after filtration). The other sample was a piece of the same film remaining after cutting out the above fragment (native sample before filtration).

Optical microscopy

The optical images were obtained using a Nikon SMZ1270 stereoscopic microscope. The native (non-treated) dry samples were studied.

Membranes were examined before and after the filtration experiments. The upper and lower surfaces of samples were studied without additional preparation.

Scanning Electron Microscopy (SEM)

Membrane morphology was studied by scanning electron microscopy using a Jeol FE-SEM JSM 6400-F instrument.

The SEM images were also obtained with the aid of (Headquarters) Thermo Fisher Scientific, Waltham, MA (Massachusetts), USA at the accelerating voltage of 1 kV.

Two types of cross-sections were studied: low-temperature cleavages and thin cuts of films.

#### 2.3.4. SEM Surface Reconstruction

The 3D visual representation of the surface was performed using the surface reconstruction technique described in detail in [38,39] on the basis of scanning electron microscope images. This technique allows one to obtain the 3D surface model of an object and is also useful for quantitative analysis of the surface morphology [40]. The technique involves acquisition of (stereo) pairs and triplets of sample images at eucentric tilts of 5° to 10° in the opposite directions from the horizontal position. The images were obtained with an SEM (Headquarters) Bruker Alicona, Raaba/Graz, Austria. at the accelerating voltage of 1 kV. The 3D surface model was constructed from the aligned images using the photogrammetric approach. The image processing and 3D surface reconstruction were performed using the MeX software (Bruker Alicona, Oettingen).

### 2.4. Transport and Separation Properties

Transport experiments were carried out for three sample types:

(i) the wet samples that were not subjected to any preliminary manipulations; (ii) the wet samples that were repeatedly washed with distilled water and conditioned in the filtration cell in an aqueous medium for 24 h; and (iii) the dry samples (the dried initially wet films) that were conditioned in a penetrant in the filtration cell for 24 h.

#### 2.4.1. Transport Properties of Porous Samples

Water and ethanol permeances of the membrane were determined by measuring fluxes (*Q*, kg·m^−2^⋅h^−1^) of penetrants (*Q_i_*) and their mixtures (*Q*_total_ = *Q*_t_) by use of the ultrafiltration cells (Millipore) with effective membrane areas equal to 2.835⋅10^−4^ or 3.14⋅10^−4^ m^2^. During the experiment, the transmembrane pressure drop in the range of 1–4 atm was maintained using compressed nitrogen.
Qi=miS·t,
where *m_i_* (kg) is the permeate quantity of the component *i*, *S* (m^2^) is the effective membrane sample area, and *t* (h) is the outflow time.

Penetrant *i* permeance (*Q_i_*/Δp) was estimated as a pressure normalized flux value.

The penetrant was transported through the membrane at various pressures (1–4 atm). When studying the response of the membrane to loads, the pressure was first increased stepwise with a step of ≈0.4 atm; then the pressure was reduced in the same way. The pressure was created using a gas inert to the penetrant (nitrogen) and measured with a manometer. At each measurement point (at a certain pressure), the permeate was weighed with the aid of an analytical balance, and the time required for transport of the permeate through the membrane was recorded using a stopwatch. 

A permeate (and penetrant) composition of each measure point was analyzed on a GS-2014 Shimadzu gas–liquid chromatograph equipped with a packed GC column Porapak Q. 

The obtained results were used to construct the following plots: the pressure dependences of fluxes of pure substances (for ethanol and water); the pressure dependences of partial fluxes during the filtration of ethanol–water mixtures; the concentration dependence of a total mixture flux; the pressure dependences of “real” and “ideal” separation factors. 

The ideal separation coefficient (α) was calculated as the ratio of individual penetrant fluxes and the real separation coefficient (f) was calculated as the ratio of the partial fluxes of separated components of mixtures.

#### 2.4.2. The Cut-Off Testing

In determination of the cut-off point, diluted (C = 5–8%) aqueous solutions of PEGs with molecular weights of 600, 400, and 200 g/mol were used. The composition of permeates was determined using refractometry. 

## 3. Results and Discussion

The mild treatment of biomaterial, which was an ascidian tunic, with aqueous solutions of alkali, made it possible to remove almost all organic substances from it, except for chemically resistant cellulose. The method described above made it possible to obtain anisotropic films of complex architecture. It can be seen from Figure 3 that the structures of the film surfaces differ significantly. The upper surface is grayish beige, with a slightly pronounced relief, and contains brown protrusions evenly distributed over the surface (Figure 3a,c). Images taken with an optical microscope at higher magnification show the parchment texture (Figure 3e). The bottom surface of the film does not contain these pronounced color protrusions. In the photographs from this side, a gray–beige texture of a dense film is visible. The cross-sectional image shows that “the outgrowths” have a well-defined shape, especially at the border with the main film. They give the impression that they are made of a different material and can be removed from the surface. However, prolonged washing of the film in aqueous and alcoholic solutions did not lead to the removal of these formations from the surface. 

According to the literature, the chemically resistant frame of the tunic *Halocynthia aurantium* should consist of a cellulose-like substance, tunicin. As mentioned in the Introduction, the substance tunicin has not been studied in full, although there is an assumption that it is a complex of cellulose with protein. According to another version, tunicin is cellulose modified with fragments of protein molecules.

The panoramic SEM microphotographs presented in Figure 4 show that the brown protrusions on the upper surface of the film have a complex architecture that requires detailed study. The film has a layered morphology; the layers consist of long nanoscale-thick fibers with voids visible between them (Figure 4a). When we do not consider the structure of the upper layer, the analogy between the morphology of this sample and the well-studied structure of BC films is suggested [34,41,42,43]. 

Surprisingly, the cross-sectional image shows that the protrusions on the upper surface of the laminated film are formations consisting mainly of the material of the upper layer. This layer, as it were, is interrupted in the region of the outgrowth, bends at the points of ruptures, and is located perpendicular to the surface. As a result, it seems that foreign formations are present on the upper surface of the film. Of interest is the study of the chemical structure of the whole film and of its surface elements. 

First of all, the question arises whether the resulting film is a cellulose-like material, whether it contains the “complex” of tunicin, described in the literature when considering ascidians [44,45]. In this regard, the anisotropic film obtained in this work was studied using IR spectroscopy in the ATR mode. The choice of the method is due to the fact that the spectra of cellulose from various natural sources are well described [46]. In addition, the ATR method makes it possible to study the structure of a film on both sides (which in our case differ significantly) [47].

### 3.1. Spectral Characteristics

According to the IR spectroscopy data, Cellkon-AKH is a cellulose sample of animal origin. The dried film was examined. The spectra of both surfaces (1 and 2) were registered, and the difference spectrum (3) was obtained subtracting spectrum (1) from spectrum (2) (Figure 5).

The spectrum of surface (1) (lower surface) contains the following bands typical of cellulose: 3400 cm^−1^ (OH stretching vibrations), 2920–2850 cm^−1^ (symmetric and antisymmetric vibrations of CH_2_ groups); the peaks at 1107, 1050, 1030, and 1000 cm^−1^ are attributed to vibrations of the glycosidic ring. In the spectrum of surface (2) (upper surface), the peaks related to cellulose are less intense, but the bands at 3290, 1640, and 1530 cm^−1^ are clearly visible. In the difference spectrum (3), these bands become even more pronounced. The presence of the maximums at 3290 cm^−1^ (NH stretching vibrations, amide A), 1640 cm^−1^ (C=O absorption, amide I), and 1530 cm^−1^ (NH bending vibrations, amide II) in the difference spectrum indicates that the upper surface is composed of proteins.

According to the data discussed above, the lower and inner layers of the studied sample consist of cellulose. Small amounts of protein molecules are present on the upper surface, which agrees with the classical concept about the chemical structure of tunicin (a cellulose-like substance, namely, a polysaccharide modified with peptide chains [32]). The literature contains no detailed information on the structure of this material. It should be noted that the amount of peptide (protein) fragments and impurities in the studied film is extremely low. The composition, structure, and distribution of these fragments on the upper surface of the film require special investigation.

As mentioned above, the layered-fibrous morphology of the studied film resembles the morphology of a BC film. The literature does not contain any information about the structure of the Cellokon-AKH sample, since this film was investigated for the first time in the present work. FTIR spectroscopy data and the analogy that can be traced in the general layered-fibrous morphology of the film with known films of BC, as well as literature data, indicate that the observed polysaccharide is cellulose. 

Therefore, an attempt was made to obtain some preliminary information about the structure of this film using X-ray diffraction analysis (Figure 6).

### 3.2. X-ray Analysis

Since it is typical for cellulose to form allotropic modifications, especially with the involvement of solvents in the structure, wet and dry samples of an anisotropic film were studied [47]. 

The X-ray diffraction patterns (reflection geometry) of two sides of the samples obtained immediately after pre-treatment (wet samples) are provided in Figure 6 (curves 1, 2). Diffractograms 3 and 4 were registered after drying these samples in air. The analysis of the obtained diffraction patterns showed that the samples consist of crystalline cellulose of the monoclinic modification Iβ [48]. The faces of the monoclinic cell of cellulose, which form the surface of microfibrils, are the (–110) and (110) planes. Microfibrils with diameters of 80–120 Å are predominantly stacked parallel to the surface to form nanosized ribbons with a rectangular cross-section and the following two-dimensional lattice parameters: a = 0.801 nm, b = 0.817 nm. The change in the ratio between intensities of the (−110) and (110) reflections in the diffraction patterns indicates that drying of the film results in the predominant orientation of the microfibril bundles along the (−110) plane parallel to the sample surface [49].

To summarize, the obtained membrane is a cellulose-based film that retains its native complex architecture. Probably, it is precisely this architecture of the cellulose base of the ascidian tunic that allows the animal to exist under high pressure conditions in the sea at depths of 50–400 m. Therefore, the film of this structure should be stable under the conditions of pressure-driven processes of transmembrane transport of liquids. The results of the studies of transport properties of these films depending on the applied pressure are discussed below. 

### 3.3. Transport Properties

The data presented above show that Cellokon-AKH is an anisotropic film with significantly different upper and bottom surfaces. If we consider this film as a membrane for pressure-driven filtration, it is not clear which elements of the film structure limit the resistance to the flow of liquid passing through it.

In such cases, the question arises regarding which layer of the membrane can be considered the upper one, from the side of which the mixture to be separated is passed under pressure. The membrane in the separation cell is placed higher with a layer, which is recognized for a particular experiment as the top one. From the side of this layer, the mixture to be separated is passed through the membrane. It is known that for an anisotropic membrane, the flux directed from the surface (1) towards the surface (2) may or may not be equal to the flux transmitted under the same conditions from the surface (2) towards the surface (1) [47].

It is important to establish whether it is possible to change the flux through the membrane by changing its position in the separating cell (namely, by changing the membrane side contacting with separated liquids). 

Moreover, it should be considered that the packing densities of polymer chains are different in wet and dry membranes. It is clear that wet and dry membranes will demonstrate different permeabilities. Obviously, less densely packed wet films should have greater permeability. However, two questions arise: will these wet films show selectivity, and how will pressure affect their properties?

#### 3.3.1. The “Wet” Film

The test results show that the Cellokon-AKH film stored in 96% aqueous ethanol (“wet” film) is a gel film. 

The “wet” film, stored in 96% ethanol, placed in the separation cell with its upper side (as in Figure 3) in contact with the liquid to be separated, and maintained at a pressure of 1 atm until the permeate flux into the receiver ceased (which was controlled by weighing the receiver), lost 69.67 wt% of its original weight. The amount of released liquid is 229.7% of the mass of dry matter.

After the membrane lost fluid and had a constant weight, the ethanol flux through the same membrane (without removal from the cell) was measured at 1 atm, which amounted to 153 kg·m^−2^·h^−1^. 

Then, the 48% solution of aqueous ethanol was passed through the membrane at 1 atm. In this case, the total penetrant flux through the membrane was 53.5 kg·m^−2^·h^−1^ at the ethanol concentration in permeate equal to 50% (the liquids were not separated). 

With an increase in the concentration of ethanol in the separated mixture, the properties of the same membrane changed significantly. When the 70% aqueous solution of ethanol transported through the membrane, the data obtained for various samples showed significant scatter. Even though the pressure remained constant and amounted to 1 atm, the flux through the membrane fluctuated; first, it reached 104.8 kg·m^−2^⋅h^−1^, then decreased down to 55.6 kg·m^−2^⋅h^−1^, while the high selectivity was observed; the permeate contained 100% of ethanol. It should be noted that the stable flux value that the membrane showed as a result at this point is almost equal to the water flux value (see below).

On the contrary, a decrease in the ethanol concentration in the separated mixture to 30% leads to a decrease in the flux through the membrane to 32–34 kg⋅m^−2^⋅h^−1^, and the concentration of ethanol in the permeate is 26–34 kg⋅m^−2^⋅h^−1^ (scatter for repeated experiments), which indicates the lack of selectivity.

With further passage of pure water through the membrane, the flux was 55.4 kg⋅m^−2^⋅h^−1^.

Thus, the above transport properties of a wet membrane obtained by successively passing ethanol, water, and their mixtures of various compositions through it at 1 atm showed that the native membrane retains a large amount of ethanol, transports more liquids, more of in them ethanol, except for a mixture containing 30% ethanol. The permeance of the membrane for water and most aqueous ethanol mixtures was ~55 kg⋅m^−2^⋅h^−1^, and only in one case (70% ethanol) the wet membrane had high selectivity. A high and generally stable level of flux against the background of a lack of selectivity indicates the presence of large transport channels in the membrane, which are characteristic of gel-like polymeric films, such as a film of BC. 

The same sample was placed in the cell with its lower side oriented toward the penetrant. The measurements were carried out under the same conditions at a pressure of 1 atm. In this case, the flux of 96% ethanol was 202.5 kg⋅m^−2^⋅h^−1^, of 30% aqueous ethanol −37.5 kg⋅m^−2^⋅h^−1^, of 48% aqueous ethanol −26.5 kg⋅m^−2^⋅h^−1^, of 70% ethanol 56.5 kg⋅m^−2^⋅h^−1^ at an ethanol concentration in the permeate of 65%, and of water 73 kg⋅m^−2^⋅h^−1^. The data arranged in a row is also provided in accordance with the sequence of measurements.

Thus, transport properties of the wet initial membrane differ significantly depending on the arrangement of the film in the separation cell. At the same time, the membrane is more permeable when it contacts the penetrants with its lower part. It is difficult to provide a simple explanation for this “behavior” of the membrane. Although it can be assumed that the polymer that forms the membrane structure enters into intermolecular interactions with the formation of a complex network of hydrogen bonds. Such interactions are characteristic of polysaccharides and, first of all, of cellulose. It is important that in films of such a complex structure such as those of Cellokon-AKH, various network polymer crosslinks can occur in different regions. It is especially difficult to explain why a wet membrane is selective only when separating a 70% aqueous mixture of ethanol. Nevertheless, such an unusual property can be associated with two factors: (1) with the features of the super-complex structure of the membrane and (2) with the specifics of its storage in ethanol.

#### 3.3.2. The “Dry” Film

The so-called “dry” membrane was obtained by drying the original membrane (n.a.) under tension. The membrane placed in the cell with its upper surface to the penetrant showed low water permeability. When the pressure was increased to 4 atm., no water appeared in the receiver. However, the total penetrant flux through the membrane at 4 atm. was 0.378 kg⋅m^−2^⋅h^−1^ (*Q*/Δp = 0.09 kg⋅m^−2^⋅h^−1^⋅atm^−1^) for the 10% aqueous ethanol solution, and it was equal to 0.64 kg⋅m^−2^⋅h^−1^ (*Q*/Δp = 0.16 kg⋅m^−2^⋅h^−1^⋅atm^−1^) for the 85% aqueous solution of ethanol. In the latter case, the concentration of ethanol in the permeate was 4%, that indicates the predominant transport of water through the membrane. Under the same conditions at 4 atm., the membrane also showed high ethanol permeability, and the flux through the membrane was 1.7 kg⋅m^−2^⋅h^−1^ (*Q*/Δp = 0.43 kg⋅m^−2^⋅h^−1^⋅atm^−1^).

Thus, an increase in the concentration of ethanol in the separated mixture leads to an increase in the flux through the membrane. However, the actual separation selectivity exhibited by the membrane is due to the preferential transport of water. This contradiction can be explained by possible structural changes in the membrane that occur after the penetration of ethanol. Apparently, small rearrangements in the system of hydrogen bonds caused by ethanol molecules are accompanied by the release of hydrophilic groups (-OH, etc.) responsible for water transport.

The membrane tested in this way was kept in the separation cell under water for 2 months. After conditioning, its selective transport properties were studied, which are presented in Table 2 (the upper side of the membrane to the feed mixture). The total flux of penetrants through the membrane increased (the results presented in the table were obtained at 2 atm.).

The data presented in Table 2 show that with an increase in the concentration of ethanol in the mixture, the total flux through the membrane also increases, while the trend towards enrichment of the permeate with ethanol persists. However, when separating a 50% aqueous ethanol solution, the membrane exhibits hydrophilicity and the permeate becomes slightly enriched with water.

#### 3.3.3. The “Freely Dried” Film

It seems that the membrane structure is sensitive to post-processing conditions. It was shown that drying the sample under tension (*“dry” film*) dramatically reduces the membrane permeability, although the initial permeability (*“wet” film*) is relatively high.

At the same time, tension prevents the contraction of the porous polymer film. Therefore, drying under milder conditions without tension (“*freely dried” film*) can lead to a greater change in the membrane structure, its compaction, as shown by X-ray diffraction (Figure 6).

An analysis of the data obtained in the study of the transport properties of a membrane freely dried under normal conditions (without tension and other external additional influences) is indicative. The membrane was placed in the cell with its upper surface to the penetrant. The permeability of the membrane for water, ethanol, and their mixtures of various concentrations increases with increasing pressure. At the same time, during the separation of ethanol–water mixtures, the partial fluxes through the membrane of both ethanol and water increase symbatically (Figure 7).

However, the ideal membrane selectivity coefficient differs significantly from the real one, which characterizes the separation of a liquid mixture (Figure 8). This correlates with the results described earlier in Section 3.3. This means that the ideal water selectivity coefficient for a given membrane is less than one, that is, the membrane is selective at a higher ethanol transport rate. At the same time, the separation factor for water of a real mixture of liquids is greater than one, that is, water is predominantly transported through the membrane (Figure 7 and Figure 8). Of course, the mechanism of transport of liquids through this membrane still needs to be rigorously investigated. It remains to be clarified what determines these transport properties—the affinity of a hydrogen-bonded cellulose film for penetrants (ethanol, water) or its ability to selectively pass penetrants in accordance with their size. Perhaps we are dealing with a mixed transport mechanism.

Particularly interesting results include the dependencies shown in Figure 9. It is shown that with increasing pressure, the permeability of a freely dried membrane monotonically increases for both ethanol (Figure 9a) and water (Figure 9b). On the example of water permeability, the “cyclic” dependence of flux on pressure is illustrated in Figure 9b. The decrease in pressure leads not only to a decrease in the flux through the membrane, but also to a decrease in the flux to the same intermediate values that corresponded to the given intermediate pressure values on the rise curve. In other words, the graph does not show a hysteresis loop characteristic of many porous membranes. This means that in the pressure range from 1 to 4 atm., the membrane undergoes only elastic deformations and there are no irreversible changes in the supramolecular structure of the polymer material. The absence of irreversible deformations in the membrane structure is confirmed by the results of morphology studies of membrane samples taken before and after filtration (Figure 3). 

In determination of the cut-off point, diluted (C = 5–8%) solutions of PEG with molecular weights of 600, 400, and 200 g/mol were used. For the solution of PEG with a molecular weight equal to 600 g/mol, no permeate flow was observed at a pressure of 4 atm. When solutions of polymers with MM of 400 or 200 g/mol were used, traces of PEG were found in the permeate. These data indicate that the cut-off point lies in the range from 400 to 600 g/mol.

Thus, the investigated Cellokon-AKH under moist initial conditions (biomaterial storage conditions) is a structurally stable gel film containing approximately 230% ethanol (storage medium) by weight of dry matter. After the removal of liquid from it under the conditions of a filtration cell and subsequent passage of ethanol, water–ethanol mixtures, and water through it, the film exhibits high permeability to all of the above penetrants. This is manifested to a greater extent if the membrane is placed in the cell upside down, that is, in contact of the lower layer with the liquids to be separated. If, however, penetrants are passed through this film under pressure in the direction from the upper surface to the lower, for the penetrant containing 70% aqueous ethanol, the membrane is selective. In connection with the tendency to exhibit separation properties under these conditions, the upper membrane side was accepted as the working one, located in all further experiments in the separation cell from the side of the mixture being separated.

After drying in air under normal conditions without additional influences, as well as under tension, the film lost its gel properties, and therefore the liquid flux through the membrane significantly decreased. Apparently, the membrane structure in these cases became denser, which correlates with the X-ray diffraction data (Figure 6). As a result, the membrane acquires a pressure-resistant structure, and no inelastic deformations are observed. However, the flows through the membrane now correspond to the level of nanofiltration processes. The same is illustrated by the filtering ability of the films at the complete cutoff of polyethylene glycol molecules 600 Da, which corresponds to nanofiltration membranes. The flux level fluctuates slightly as a result of membrane conditioning. Although long-term exposure to water leads to some increase in permeability, its reverse transition to the gel-film state does not occur. Such properties are characteristic of known BC membranes, in which the formed network of hydrogen bonds prevents the penetration of water into the space between the polymer chains.

Summarizing the data presented above, it can be assumed that the anisotropic membrane under study, with an overall extremely complex structure, should have nanosized pores and through channels, which is typical for nanofiltration membranes.

Its resistance to pressure increase is high and no general structural deformations are observed (Figure 3). However, visual changes occur in the area of the protrusions on the upper surface of the membrane (Figure 10c). It remains unclear whether significant deformation of the cellulose framework occurs in these areas. To what extent do these outgrowths protrude above the surface of the film?

To clarify the structural features of the membrane, an attempt was made to study it using high-resolution SEM.

### 3.4. SEM Analysis of Prepared Membranes

High-resolution SEM images of membrane surfaces in Figure 10 are similar to those shown in Figure 4.

From the upper side of the membrane, no pores are visible on the main surface (Figure 4 and Figure 10). At the same time, in the area of outgrowths evenly distributed on the surface, a hierarchical porous structure is visible (Figure 10b and Figure 11a). Figure 10c,d shows that after filtering, the contours of these complex formations are slightly blurred, but the pores in the center of the outgrowths remain visible.

The consideration of SEM images of surface structures allows (Figure 3, Figure 4 and Figure 10c,d) to suggest that the protein component may not be a part of the polysaccharide, but be located on the surface of the sample. There is an assumption that it is in this area that peptide fragments grow, as shown in Figure 3, in a brown–burgundy color, the detection of which was established by IR spectroscopy (Figure 5). Such areas may have a less rigid structure subject to deformation under pressure. This assumption requires additional research, including an analysis of the chemical composition in various parts of the surface (mapping).

An image in Figure 10a shows the film from the side of the bottom surface. It clearly shows the fibrous structure of this part of the membrane, in which there are porous areas. However, the pores cannot be what we are looking for. They are too large. 

Figure 11 shows the results of a more thorough study of the membrane surfaces in order to determine its pore structure.

The local changes in membrane surface morphology before and after filtration were studied using low-voltage scanning electron microscopy (LVSEM) [50]. Figure 11a,c shows that the membrane surface before filtration is a network of intertwined cellulose fibrils, with both denser and looser regions present on the lower surface compared to the completely uniform morphology of the upper surface. The average pore size for the upper surface is about 100–150 nm, for the lower surface it is about 60–100 nm. After filtration, the nature of the surface morphology changes significantly. Figure 11d shows that the underside of the membrane becomes friable with micron-sized pores. On the upper surface (between surface formations) one can observe fibril bunches with a thickness of 0.1–0.5 µm and individual fibrils are practically indistinguishable.

However, the images in Figure 4b and Figure 10d,f showing the cross-section of the membrane (Figure 4b), its cut surface in the outgrowth area (Figure 10e), as well as the low-temperature cleavage surface in the outgrowth area, illustrate that the main material of the membrane is cellulose, the fibers of which are located in the film plane. They bend in the outgrowth area perpendicular to the membrane surface. 

Due to the high level of complexity of the outgrowth-like formations of interest to us on the surface, it is extremely difficult to establish their size and internal structure. In such cases, resort to methods of structural modeling.

### 3.5. Sample Surface Structure Reconstruction

The result of sample analysis using a scanning electron microscope is a projection of a three-dimensional (3D) surface structure onto a two-dimensional image plane. However, quantitative information about the depth of the sample (i.e., features of the object along the direction of the electron beam) may be lost. To extract this information, it is necessary to apply the special method for creating a three-dimensional image, using the phenomenon of the stereo effect (2. in Appendix A).

The photogrammetry technique can be used for 3D reconstruction of SEM images. SEM is one of the most commonly used surface imaging methods. SEM has a large depth of focus and high resolution (up to 1 nm) and is ideal for stereo imaging and 3D surface reconstruction. The technique for obtaining a stereo pair of images in SEM is based on repeated image acquisition of the same object tilted at different angles (1–20°) with respect to the electron probe. The angle can be changed mechanically by tilting the sample using a goniometer or by tilting the electron beam itself with a fixed sample (if there is a special deflecting device in the microscope). Computational technologies make it possible to numerically reconstruct the sample surface, i.e., to generate its three-dimensional model, and to perform quantitative analysis of the generated surface model.

The results of the protrusion area surface 3D reconstruction are presented in Figure 12. In the presented model, color variations correspond to the relief height relative to the base plane (the scale is provided in the figure). The protrusion height measurements were performed by cross-sectional survey. Figure 12c shows the protrusion surface cross-section and height measurement. The height of the structure is up to 127 µm.

The presented method, tested by us to obtain information about the structure of surface formations in a polymer film of complex architecture, is supposed to be further used in modeling and visualizing its internal surfaces.

## 4. Conclusions

A new film material of hierarchical architecture isolated using a specially developed method from the ascidian Halocynthia aurantium was studied. The use of mild conditions for the isolation of a film based on tunicin from the sea peach tunic made it possible to preserve the structural and morphological features of this material, thought out and realized by nature when creating the underwater animal world. The study and application of a film that retained its natural structure (bionics) made it possible to approach new methods for solving well-known problems. These are the technical problems, such as the development of environmentally friendly, chemically resistant, and resistant to external loads (high pressure) filter materials or the development of marine resources, as well as wider problems, such as the structure of marine fauna, which, in terms of the structure of tunicates, has not been studied as deeply as in this work.

Based on the analysis of the structure and transport properties of the anisotropic Cellokon-AKH film obtained from the AnC of the *Halocynthia aurantium*, its use as a membrane for nanofiltration of substances with a molecular weight of more than 600 Da in aqueous and alcoholic media is substantiated.

The data obtained showed significant differences in the structure and morphology of the upper and lower surfaces of the layered film membrane.

It is shown that a set of unique properties of the membrane, including structural stability at elevated pressures, is determined by its extremely complex structure, “thought-out nature” when creating the mantle of deep-sea animals of the *Halocynthia aurantium* class. The structure of the most complexly organized sections of the upper layer of the membrane was determined and analyzed. The surface model was calculated from merged high-resolution scanning electron microscopy images based on the photogrammetric approach, which made it possible to estimate the sizes of porous surface areas with a hierarchical architecture. However, the studies of the *Halocynthia aurantium* tunic presented in this paper provide only general ideas about its structure. Thus, a logical continuation of this work, in our opinion, should be a detailed study of not only the mechanism of mass transfer through Cellokon-AKH, but also the features of its chemical composition simultaneously with penetration into the inner layers of the membrane in order to establish their structure.

## Figures and Tables

**Figure 1 membranes-12-00975-f001:**
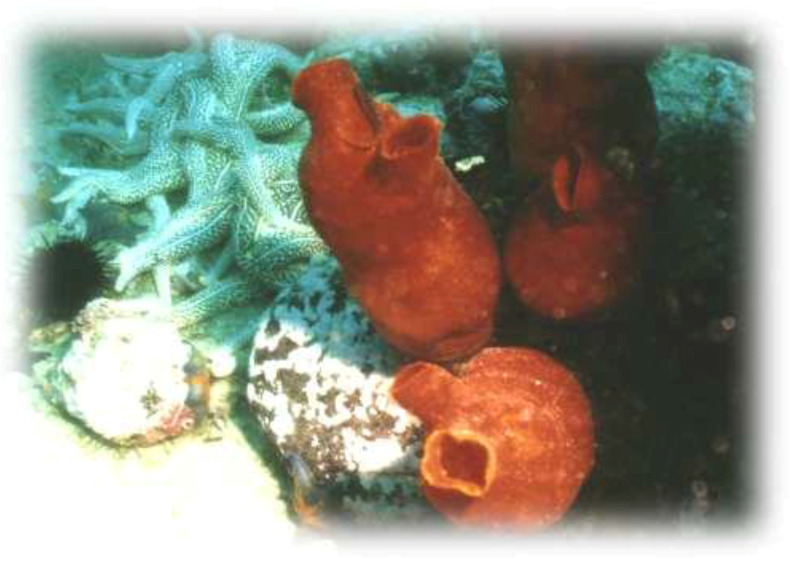
*Halocynthia aurantium*, commonly known as the sea peach (underwater photography, the image presented on the website of the Far East Geological Institute of the Far Eastern Branch of the Russian Academy of Sciences [24]).

**Figure 2 membranes-12-00975-f002:**
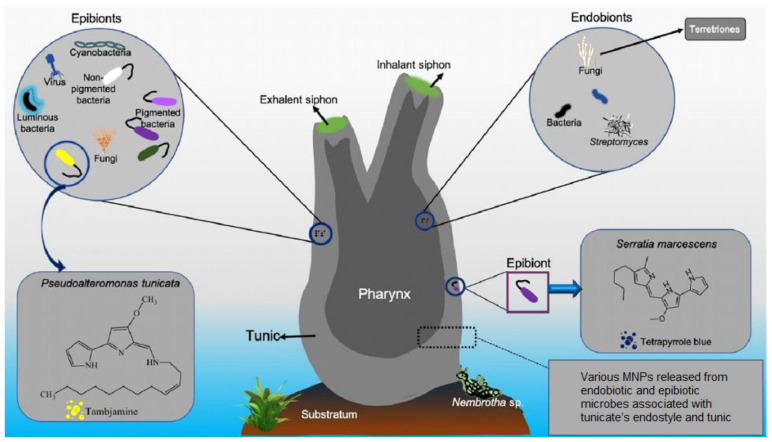
Tunicate (*Halocynthia aurantium*) and tunicate-associated symbionts [25].

**Figure 3 membranes-12-00975-f003:**
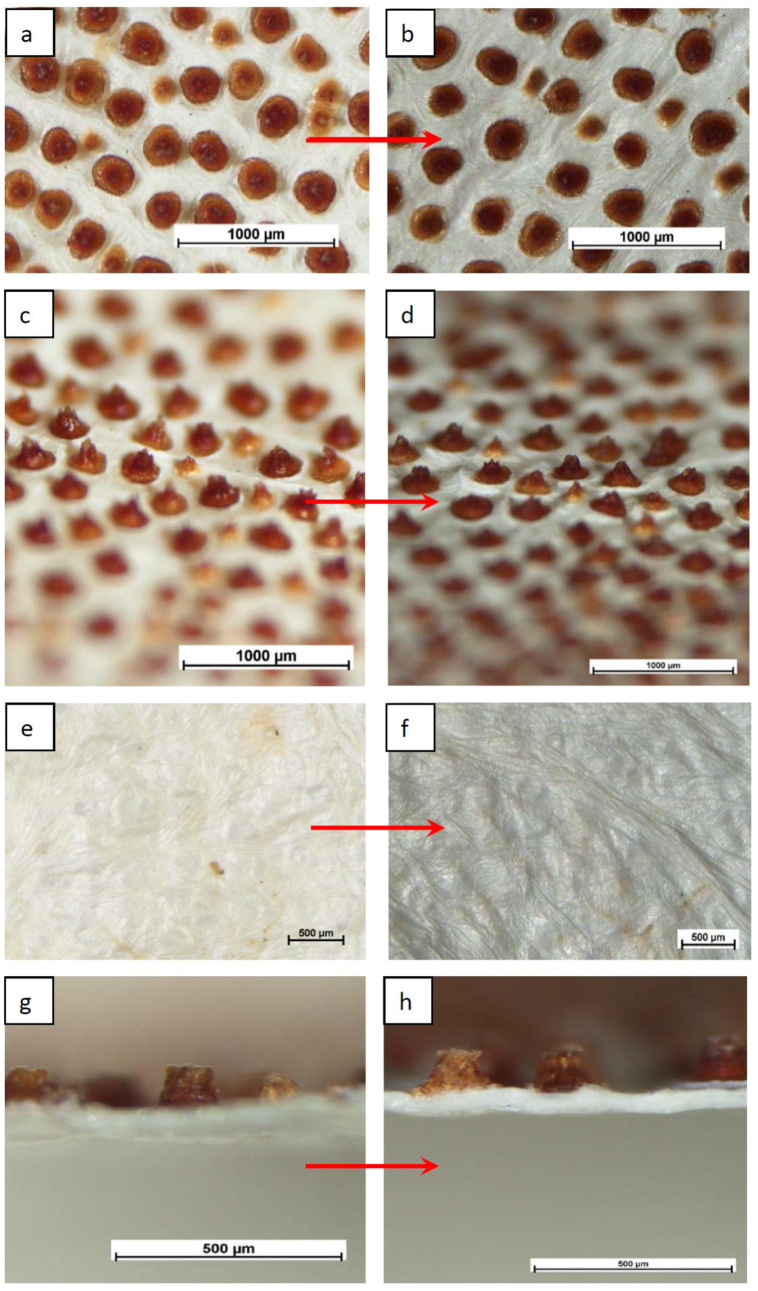
Optic microscope images of Cellokon-AKH film: (**a**–**d**) upper surface; (**e**,**f**) bottom surface; (**g**,**h**) cross-section.

**Figure 4 membranes-12-00975-f004:**
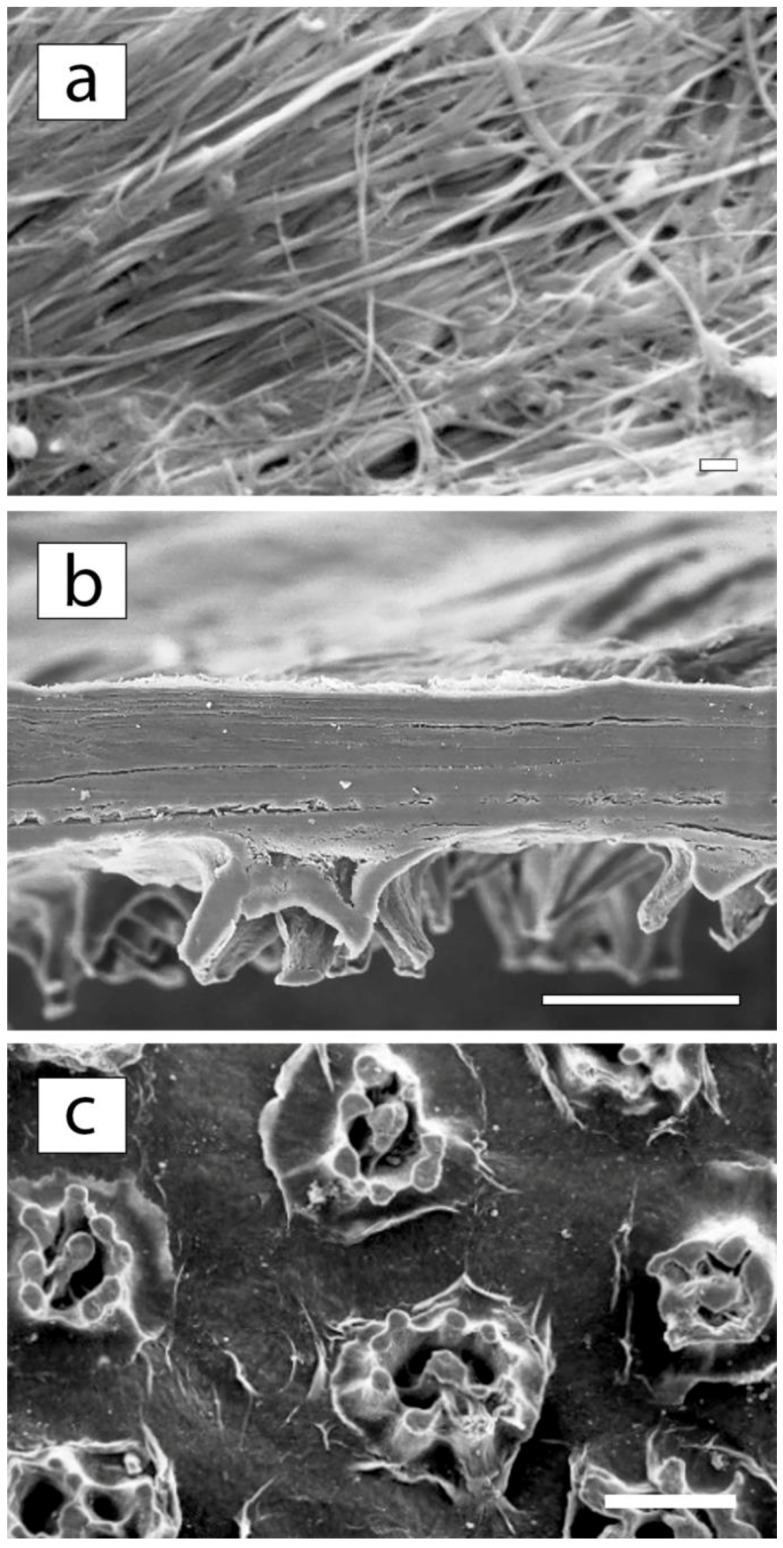
The SEM micrographs: (**a**) the bottom surface (×6000; bar—1 micron), (**b**) the cross-section (×300; bar—100 microns), and (**c**) the upper surface (×200; bar—100 microns) of the Cellokon-AKH film.

**Figure 5 membranes-12-00975-f005:**
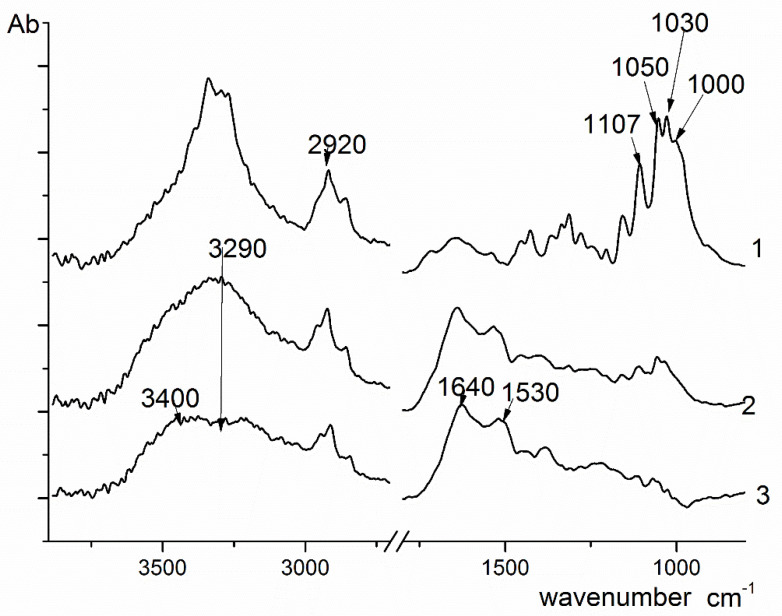
The FTIR spectra of lower (1) and upper (2) surfaces of the Cellokon-AKH film; the difference spectrum (3) obtained by subtracting spectrum (1) from spectrum (2).

**Figure 6 membranes-12-00975-f006:**
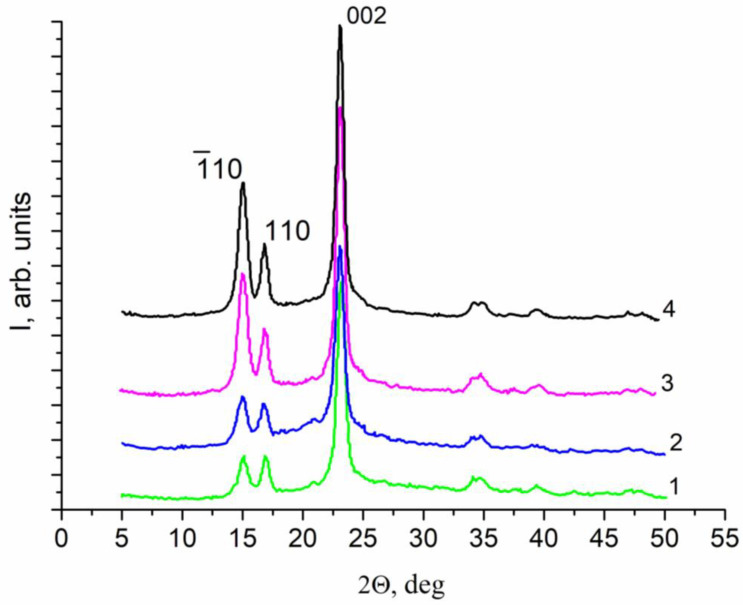
X-ray diffraction patterns of Cellokon-AKH samples. Curves: 1—bottom side of the wet film; 2—top side of the wet film; 3—top side of the dry film; 4—bottom side of the dry film.

**Figure 7 membranes-12-00975-f007:**
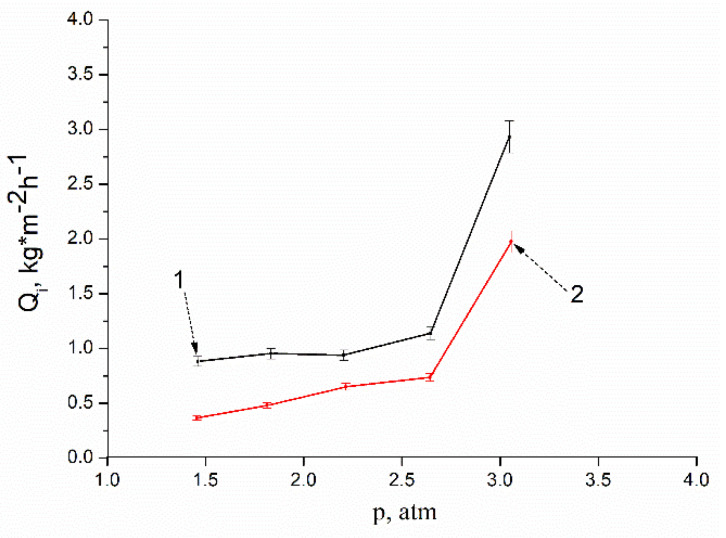
Pressure dependences of the partial fluxes Q_i_ (kg·m^−2^·h^−1^): (1) water; (2) ethanol.

**Figure 8 membranes-12-00975-f008:**
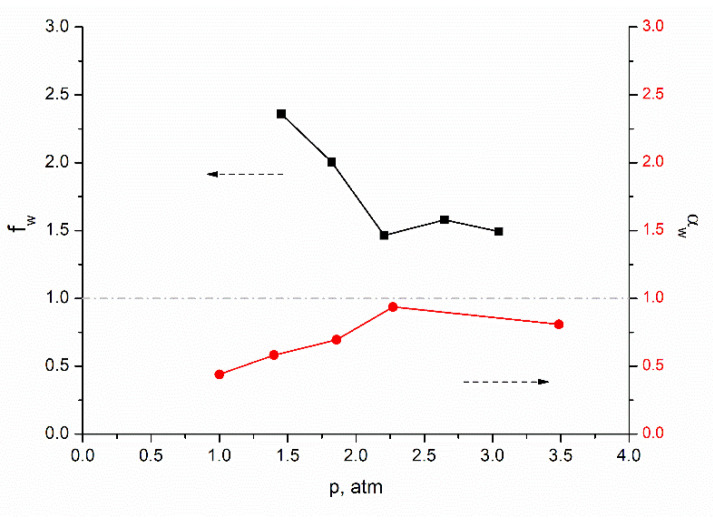
Pressure dependences of the ideal selectivity coefficient α_w_ and the real selectivity coefficient f_w_.

**Figure 9 membranes-12-00975-f009:**
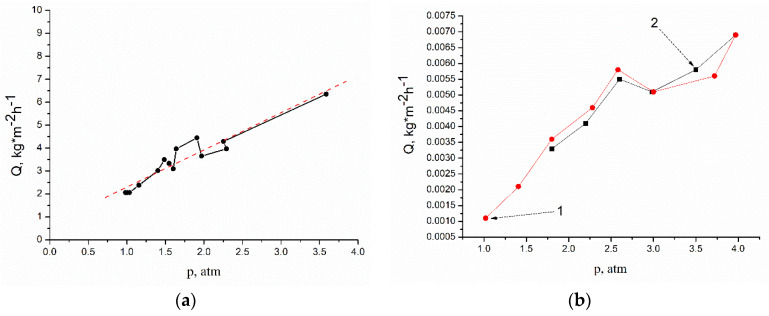
Pressure dependences of Q (kg·m^−2^·h^−1^) for ethanol (**a**) and for water (**b**); (1) increasing pressure during the experiment; (2) the experiment with decreasing pressure.

**Figure 10 membranes-12-00975-f010:**
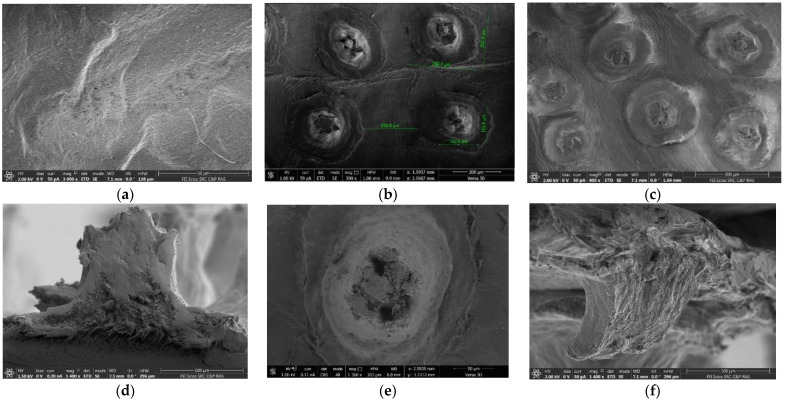
The SEM micrographs of bottom (**a**) and upper (**b**,**c**) membrane surfaces and individual structural elements localized on the upper (**d**–**f**) surfaces of the film, its cross-section (**d**,**f**), and slice parallel to the membrane plane (**e**). The SEM micrographs in (**a**,**b**,**e**,**f**) belong to the membrane before the baromembrane process; in (**c**,**d**) after it.

**Figure 11 membranes-12-00975-f011:**
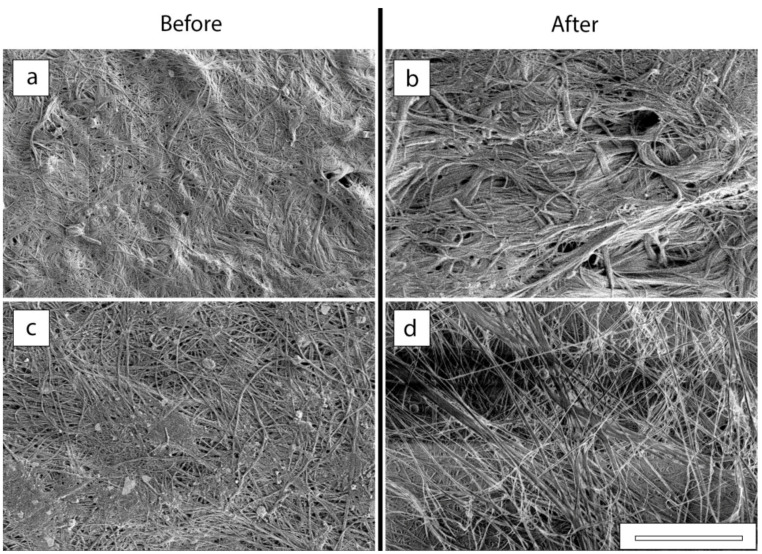
The SEM images of the upper (**a**,**b**) and lower (**c**,**d**) surfaces of the films before (**a**,**c**) and after (**b**,**d**) perforation, respectively; bar—10 microns.

**Figure 12 membranes-12-00975-f012:**
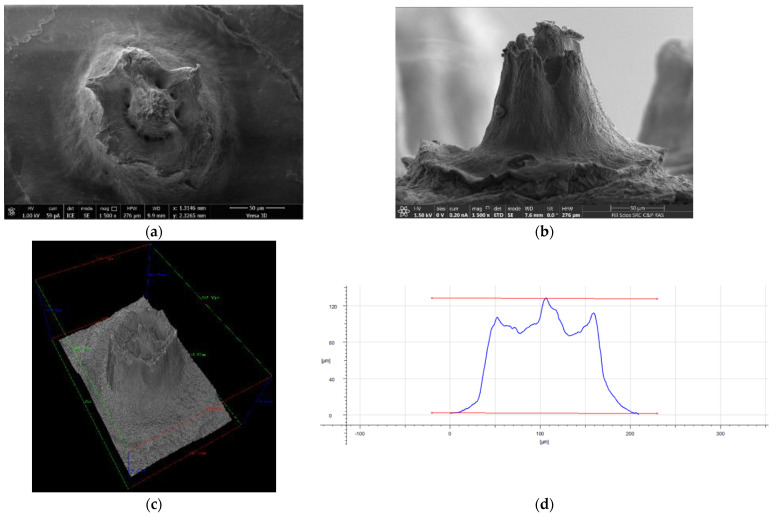
The SEM images (**a**,**b**), 3D surface models (**c**,**e**), and surface model cross-section (profile) with height measurement (**d**) of a fragment of the upper surface membrane structure (protrusion).

**Table 1 membranes-12-00975-t001:** Chemical profiles from body parts and fluids of tunicate species [25].

Body Component	Chemical Compound	Function	Application
Tunic (*Ascidia* sp*., Ciona intestinalis*, Halocynthia and *Styela plicata*)	Tunicin (cellulose)	Protection	Material cellulose
Hemocytes (*Halocynthia papillosa*)	Halocyntin and papillosin		Antimicrobial
Hemocytes (*Halocynthia aurantium*) Gonad (unknown sp.)	Halocidin GnRH-2 peptide	Pheromone-like function	Antimicrobial

**Table 2 membranes-12-00975-t002:** Selective transport properties of the membrane (the upper side of the membrane contacts with the separated mixture, Δp = 2 atm).

N	Feed (%/%), EtOH/HOH	Permeate (%/%), EtOH/HOH	*Q*, kg⋅m^−2^⋅h^−1^	*Q*/Δp, kg⋅m^−2^⋅h^−1^⋅atm^−1^
1	100/0	100/0	0.74 ± 0.04	0.37 ± 0.02
3	85/15	91/09	0.69 ± 0.03	0.345 ± 0.017
5	50/50	44/56	0.29 ± 0.01	0.145 ± 0.007
7	15/85	42/58	0.22 ± 0.01	0.110 ± 0.005
9	0/100	0/100	0.016 ± 0.001	0.0080 ± 0.0004

## Data Availability

Not Applicable.

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
