# Peer review of "Animal Cellulose with Hierarchical Structure Isolated from Halocynthia aurantium Tunic as the Basis for High-Performance Pressure-Resistant Nanofiltration Membrane"

_membranes, 2022, doi:10.3390/membranes12100975_

Round 1
Reviewer 1 Report
The work seems interesting regarding its results and suggested production methods. However, there are some minors points that the authors should clarify before acceptance. The minors points have been stated as follows:
1. More data should be presented in the abstract to emphasize the superiority of the work.
2. with respect to the analysis through SEM, write in more detail the methodology. Some essential properties of membrane is not provided, including the dimensions of pores and fibers and roughness, grooves, etc (e.g. SEM)
3. It is clear that the dimensions of the pores and fibers of Cellokon-AKH membrane are in the order of microns. Therefore, replace "nanoporous" with "microporous" and "Nanofiltration with "microfiltration"
4. Some figures need more interpretation and analysis, such as figure 7
5. some caption of figures should be rearranged. It is not scientific, such as fig 7a (at xxx atm), fig7b (at xxx EtOH concentration), replace "concentration" with "concentration of ethanol " , fig 4 ( a , b and c re-write in caption)
6. Some figures need more interpretation and analysis, such as figure 7
7. Y axis need to be determined in some fig, such as figure 5 (wave number) and 6 (2θ)
8. The fig1 is not necessary, it can be deleted
9. Some essential properties of the Cellokon-AKH membrane are not provided, including the thermal properties (TGA), mechanical properties (Tensile testing, Elongation at break and Modulus of Elasticity), water vapor permeability (WVP) and Transparency. The authors need to provide convincing evidence to support their conclusion.
10. the results presented in the table 1 were obtained at 2 atm, why this point (2atm) and at 4atm not presented in fig 7b
11. ) The manuscript should be re-writing and proofread by a native speaker.
Author Response
- Response to first reviewer
It is very pleasant that this "atypical" study aroused the interest of the reviewer.
We have tried to answer the questions posed.
In addition, we have carefully revised the article in order to improve it. |
The work seems interesting regarding its results and suggested production methods. However, there are some minors points that the authors should clarify before acceptance. The minors points have been stated as follows:
- More data should be presented in the abstract to emphasize the superiority of the work.
Response: The abstract has been re-written.
- with respect to the analysis through SEM, write in more detail the methodology. Some essential properties of membrane is not provided, including the dimensions of pores and fibers and roughness, grooves, etc (e.g. SEM)
Response: clarifications were made to the methodology, microphotographs with descriptions were added.
- It is clear that the dimensions of the pores and fibers of Cellokon-AKH membrane are in the order of microns. Therefore, replace "nanoporous" with "microporous" and "Nanofiltration with "microfiltration"
Response: Since we are studying this material in its various states, a detailed description of the filteration properties has been added into the article and appropriate changes have been made to the text.
- Some figures need more interpretation and analysis, such as figure 7
Response: Significant changes have been made to the text of the article.
- some caption of figures should be rearranged. It is not scientific, such as fig 7a (at xxx atm), fig7b (at xxx EtOH concentration), replace "concentration" with "concentration of ethanol " , fig 4 ( a , b and c re-write in caption)
Response: Changes have been made to the text of the article
- Some figures need more interpretation and analysis, such as figure 7
Response: Changes have been made to the text of the article
- Y axis need to be determined in some fig, such as figure 5 (wave number) and 6 (2θ)
Response: Changes have been made to the text of the article
- The fig1 is not necessary, it can be deleted
Response: The animal from which the biomaterial is used is little known to the general reader. To improve understanding of how the membrane is prepared, we decided to place this figure. We would like to leave it in the article.
- Some essential properties of the Cellokon-AKH membrane are not provided, including the thermal properties (TGA), mechanical properties (Tensile testing, Elongation at break and Modulus of Elasticity), water vapor permeability (WVP) and Transparency. The authors need to provide convincing evidence to support their conclusion.
Response: The membrane presented in the article is a completely new material. There is some analogy in its properties and properties of bacterial cellulose. In the corrected version of the article, its states are described in more detail (gel film, dry dense film, ..). Due to the fact that this is a very structurally complex material, additional methods are needed to study it. We are working on this now. Let's try to apply the methods commonly used by us (the study of thermophysical properties and mechanical properties) to study this object. But this is a separate study.
- the results presented in the table 1 were obtained at 2 atm, why this point (2atm) and at 4atm not presented in fig 7b
Response: The article has been substantially rewritten and more clearly organized. The current version of the article introduced subsections, while the previous unsuccessful version had continuous text. The new version of the article emphasizes that the data in Table 2 and Fig. 7b correspond to different states of the membrane (samples with different prehistory).
- ) The manuscript should be re-writing and proofread by a native speaker.
Response: The manuscript has been re-written.
Reviewer 2 Report
The current manuscript describes the “Animal Cellulose with Hierarchical Structure Isolated from Halocynthia aurantium Tunic as the Basis for High Performance Pressure Resistant Nanofiltration Membrane”. In this study, the structure and transport properties of the nanoporous Cellokon-AKH membrane based on animal cellulose obtained from ascidian tunic were studied. The manuscript needs to reconstruct by the authors, and some of the explanations may need further illustration. One important point is that the abstract and the introduction have to be revised, the authors have written it too general. I consider that the scientific discussion of this manuscript needs to be enhanced with proper justifications to be published in this journal. Also, the authors should consider critically these comments to improve the quality of the work while revising the manuscript.
- The abstract needs to be revised and add the main finding in the abstract.
- Title is not appropriate, revised it.
- The introduction is too general, it is too lengthy. In the introduction, the authors have to look for the proper citations on a related topic for the particular discussion. Avoid the old reference and add recent citations related to high-performance nanofiltration membranes like Chemical Engineering Journal Volume 446, 2022, 137303; Journal of Membrane Science Volume 609, 2020, 118212 Membranes 12 (2022) 768. J. Environ. Chem. Eng. 10 (2022) 108109. etc.
- Materials details should be there in section 2.1
- In this article, the reviewer has an important concern is authors have to write section 2.2. Membrane preparation in detail.
- Equations and equations numbers should be in the proper way.
- What exact information the authors received need to explain in section 3 about the SEM images.
- Labelling should be there on FTIR peaks.
- Section 3.3 Transport properties, the texted values are completely mismatched with the figure values so the authors have to look at this section carefully and revised it completely.
- Use the error bar for all experimental data calculations.
- All membranes need to be characterized.
- The fluxes have to be calculated and justified properly.
- The conclusion looks like the abstract so the authors have to look at it and revised and highlight the main work in this section.
- Based on SEM characterization what authors want to claim needs to explain in detail with proper justifications.
- Check the whole manuscript carefully as there is no proper space between numbers and units.
- Grammar should be checked and revised throughout the whole manuscript.
Author Response
- Reply to second reviewer
The authors paid great attention to the comments of the reviewer. The manuscript has been substantially revised. Additional results are given and more detailed explanations are given also.
The current manuscript describes the “Animal Cellulose with Hierarchical Structure Isolated from Halocynthia aurantium Tunic as the Basis for High Performance Pressure Resistant Nanofiltration Membrane”. In this study, the structure and transport properties of the nanoporous Cellokon-AKH membrane based on animal cellulose obtained from ascidian tunic were studied. The manuscript needs to reconstruct by the authors, and some of the explanations may need further illustration. One important point is that the abstract and the introduction have to be revised, the authors have written it too general. I consider that the scientific discussion of this manuscript needs to be enhanced with proper justifications to be published in this journal. Also, the authors should consider critically these comments to improve the quality of the work while revising the manuscript.
- The abstract needs to be revised and add the main finding in the abstract.
Response: The abstract has been re-written.
- Title is not appropriate, revised it.
Response: An alternative version of the title of the article is proposed (at the discretion of the editors)
- The introduction is too general, it is too lengthy. In the introduction, the authors have to look for the proper citations on a related topic for the particular discussion. Avoid the old reference and add recent citations related to high-performance nanofiltration membranes like Chemical Engineering Journal Volume 446, 2022, 137303; Journal of Membrane Science Volume 609, 2020, 118212 Membranes 12 (2022) 768. J. Environ. Chem. Eng. 10 (2022) 108109. etc.
Response: We carefully revised the article to improve it, added additional references.
- Materials details should be there in section 2.1
Response: Additional information has been added to the article.
- In this article, the reviewer has an important concern is authors have to write section 2.2. Membrane preparation in detail.
Response: Additional information has been added to the article.
- Equations and equations numbers should be in the proper way.
Response: corrections made
- What exact information the authors received need to explain in section 3 about the SEM images.
Response: Additional information has been added to the article.
- Labelling should be there on FTIR peaks.
Response: Image has been improved
- Section 3.3 Transport properties, the texted values are completely mismatched with the figure values so the authors have to look at this section carefully and revised it completely.
Response: The manuscript has been re-written.
- Use the error bar for all experimental data calculations.
Response The manuscript has been re-written.
- All membranes need to be characterized.
Response The manuscript has been re-written.
- The fluxes have to be calculated and justified properly.
Response: The manuscript has been re-written.
- The conclusion looks like the abstract so the authors have to look at it and revised and highlight the main work in this section.
Response: Significant changes have been made to the text of the article.
- Based on SEM characterization what authors want to claim needs to explain in detail with proper justifications.
Response: The manuscript contains detailed explanations and additional information on the SEM with additional images.Check the whole manuscript carefully as there is no proper space between numbers and units.
- Grammar should be checked and revised throughout the whole manuscript.
Response: The manuscript has been re-written.
Round 2
Reviewer 2 Report
Authors have significantly solved the errors so the manuscript can be accepted.